# ROS-HPL: Robotic Object Search with Hierarchical Policy Learning and Intrinsic-Extrinsic Modeling

## Abstract

Despite significant progress in Robotic Object Search (ROS) over the recent years with deep reinforcement learning based approaches, the sparsity issue in reward setting as well as the lack of interpretability of the previous ROS approaches leave much to be desired. We present a novel policy learning approach for ROS, based on a hierarchical and interpretable modeling with intrinsic/extrinsic reward setting, to tackle these two challenges. More specifically, we train the low-level policy by deliberating between an action that achieves an immediate sub-goal and the one that is better suited for achieving the final goal. We also introduce a new evaluation metric, namely the extrinsic reward, as a harmonic measure of the object search success rate and the average steps taken. Experiments conducted with multiple settings on the House3D environment validate and show that the intelligent agent, trained with our model, can achieve a better object search performance (higher success rate with lower average steps, measured by SPL: Success weighted by inverse Path Length). In addition, we conduct studies w.r.t. the parameter that controls the weighted overall reward from intrinsic and extrinsic components. The results suggest it is critical to devise a proper trade-off strategy to perform the object search well. [1]

## 1 Introduction

Robotic Object Search (ROS) is a task where an intelligent agent (a.k.a. robot) is expected to take reasonable steps to approach the user-specified object in an unknown indoor environment. ROS is an essential capability for assistant robots and could serve as the enabling step for other tasks, such as the Embodied Question Answering (Das et al., 2018a). Recently, (deep) reinforcement learning (RL) has demonstrated its power at enabling robots with autonomous behaviors (Arulkumaran et al., 2017), such as navigating over an unknown environment (Mirowski et al., 2016; Zhu et al., 2017), manipulating objects with robot's end effectors (Gu et al., 2017; Popov et al., 2017; Rajeswaran et al., 2017), and motion planning (Chen et al., 2017; Everett et al., 2018). Under the RL setting, a robot learns behavioral policy by maximizing the expected rewards that are estimated by interacting with the environment physically and/or virtually. The estimated rewards serve as the reinforcement signals for robot to update its policy.

A well-known challenge to train agent to perform ROS with RL is the sparse reward issue, due to the fact that the environment and/or the location of the target object are typically unknown. With well-defined reward functions, such as the ones in Atari games (Mnih et al., 2015), the agents are shown to achieve extremely promising performance. However, it is a well-known challenge to define the reward function under the real-world scenarios(Abbeel & Ng, 2004). Typically, for the real-world applications such as object search or target-driven visual navigation, prior research prefers to construct the reward function in terms of the distance between the robot's current location and the target location under the assumption that the full information of the environment is known (Mousavian et al., 2018; Wang et al., 2018b;a). Given an unknown environment, a straightforward way is to set a high reward when robot reaches the final goal state while at all other intermediate states, the reward is either zero or with a small negative value (Zhu et al., 2017). More recently, Ye et al. (2018b)

---

[1] Our code and models can be found in `https://XXX.XXX/XXX`.

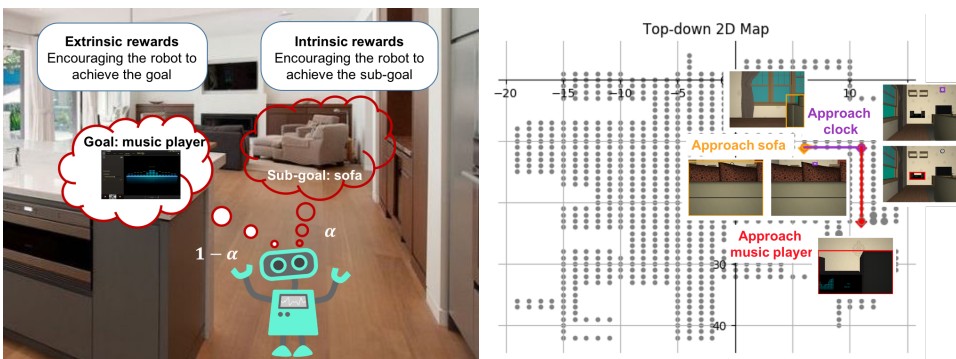

Figure 1: Left: An illustration of the *ROS-HPL* framework. The robot puts $\alpha$ of its effort in maximizing the intrinsic rewards and achieving the sub-goal, and $1 - \alpha$ of its effort in maximizing the extrinsic rewards and achieving the goal. Right: Trajectory generated from our method ($\alpha = 0.8$) for searching the target object *music player* under our SETTING-STSE.

presented a relatively denser reward function which is based on the bounding box of the target object from robot's detection system, the reward is still not defined among the situations that the target object is not detected. Thus, learning in a sparse reward setting (reward is only defined for a small subset of states) is a fundamental challenge for RL agents to perform ROS well, especially dealing with real-world scenarios and in complex environments.

To overcome the sparsity issue in reward settings, hierarchical RL has shown its superiority under the sparse reward setting (Kulkarni et al., 2016; Le et al., 2018; Levy et al., 2018). It aims to learn multiple layers of policies, in which the higher layer breaks down the task into several easier sub-tasks and proposes corresponding sub-goals for the lower layer to achieve. This kind of hierarchical policy is biologically plausible and intuitive. For human beings, when asked to accomplish a complicated task, we rarely plan over the atomic steps directly towards the target goal. Instead, we plan through a sequence of semantic meaningful sub-goals. Motivated by this observation, we put forward a novel two-layer hierarchical policy learning paradigm to deal with the sparse reward challenge.

One way to contruct the hierarchical policy learning is by adopting the option-critic framework (Bacon et al., 2017) to generate the sub-goals automatically. By specifying the number of desired sub-tasks or sub-goals, the option-critic framework can automatically abstract the whole action sequences into the specified number of the sub-goals. However, the automatically generated sub-goals typically lack semantically meaningful senses, thus are difficult for humans to understand and hurts the overall system's interpretability.

In the other way of constructing the hierarchical policy learning, as shown in work (Kulkarni et al., 2016; Das et al., 2018b), the sub-goal space could be designated from human knowledge with semantic meaningful terms. Constructing the hierarchy following human knowledge is intuitive and interpretable. Whenever the high-level layer proposes a sub-goal from the sub-goal space, the low-level layer then aims to achieve it without considering the overall goal again. However, although their results are promising for certain tasks, it is worth to mention that there is no guarantee the prior human knowledge is always perfect. The low-level layer policy which focuses only on achieving the sub-goal proposed by the high-level layer policy without keeping the final goal in mind, may drift away from being ultimately successful. This is equivalent to the scenario that without the long-term goal to provide human beings a clear direction and focus down the road, our short-term achievements are not likely to add up to something substantial.

Taking the advantages from these two ways, we put forward a system with a higher level of generalization capability upon previous work (especially, Bacon et al. (2017) and Kulkarni et al. (2016)) and present a novel framework dubbed as *ROS-HPL*: Robotic Object Search with Hierarchical Policy Learning and Intrinsic-Extrinsic Modeling, to mitigate the aforementioned challenges (see Fig. 1 left). Our contributions could be summarized in three-fold. First, we fully utilize the prior knowledge provided by human to make the hierarchy interpretable. Our high-level layer plans over a sub-goal space from human knowledge in order to achieve the final goal. Second, when a sub-goal

is proposed by the high-level layer, the low-level layer learns the action policy by deliberating between achieving the immediate sub-goal and the action that is helpful for achieving the final goal. Third, we build our framework with a novel intrinsic and extrinsic rewards setting. Though the rewards in this paper are designed for object search, the method itself is general and can be applied to other tasks. To validate *ROS-HPL*, we conduct extensive sets of experiments on the House3D (Wu et al., 2018) simulation environment, and report the observed results that validate the efficiency and efficacy of the proposed system over other state-of-the-art systems for object search.

## 2 OUR APPROACH

First, we define the robotic object search (ROS) task. Formally speaking, when a target object is specified and provided with an image, the robot is asked to search and approach the object from its random starting position. The RGB image from the robot's on-board camera is the only source of information for decision making. None of the environment information, such as the map of the environment or the location of the target object could be accessed. Once the area of the target object in the robot's viewpoint (the image captured by its camera) is larger than a predefined threshold, the agent stops and we consider it as a success. In this work, we present a novel two-layer hierarchical policy for the robot to perform the object search task, motivated by how human beings typically conduct object search. In the following sections, we first describe the hierarchy of policies. Then we introduce two kinds of reward functions, i.e. extrinsic rewards and intrinsic rewards, and we make use of these two reward functions to formulate the solution. Finally, we describe the network architecture adopted for learning the two-layer hierarchical policy.

### 2.1 HIERARCHY OF POLICIES

Our hierarchical policy has two levels, a high-level policy $\pi_h$ and a low-level policy $\pi_l$. At time step $t$, the robot takes the image captured by its camera as the current state $s_t$. Given a target object or goal $g$, the high-level layer proposes a sub-goal $sg_t \sim \pi_h(sg|s_t, g)$ and the low-level layer takes over the control. The low-level layer then draws an atomic action $a_t \sim \pi_l(a|s_t, g, sg_t)$ to perform. The robot will receive a new image/state $s_{t+1}$. The low-level layer repeats $N_t$ times till 1) the low-level layer terminates itself achieving the termination signal $term(s_{t+N_t}, g, sg_t)$; 2) the low-level layer achieves the sub-goal $sg_t$. Either way, the low-level layer terminates at state $s_{t+N_t}$, and then returns the control back to the high-level layer, and the high-level layer proposes another sub-goal. This process repeats until 1) the goal $g$ is achieved, i.e. the robot finds the target object successfully; 2) a predefined maximum number of atomic actions has been performed.

For ROS, we define the sub-goal space as {*approach obj*|*obj is visible in the robot's current view*}. We argue three reasons for the sub-goal space definition, a) approaching an object that shows in the robot's view is a more general and relatively trainable task shown by Ye et al. (2018a). It also aligns well with the goal of the hierarchical reinforcement learning by breaking down the task into several easier sub-tasks; b) approaching a related object may increase the probability of seeing the target object. As soon as the target object is captured in the robot's current view, the task becomes an object approaching task; c) as also suggested by Kulkarni et al. (2016), specifying sub-goals over entities and relations can provide an efficient space for exploration in a complex environment. Moreover, in case there is no object visible in the robot's current view, we supplement one additional candidate sub-goal: non-goal-driven exploration. The atomic action space for the low-level layer is adopted with navigation in mind, namely {*move forward / backward / left / right, turn left / right*}.

### 2.2 EXTRINSIC REWARDS AND INTRINSIC REWARDS

We define two kinds of reward functions. The extrinsic rewards $r^e$ are defined for our object search task, thus are goal dependent. Further, we also introduce the intrinsic rewards $r^i$ for the low-level sub-tasks. The intrinsic rewards are hereby sub-goal dependent. We specify the two reward functions respectively as follows.

**Extrinsic rewards $r^e$.** Without loss of generality, to encourage the robot to finish the object search task, we provide a high extrinsic reward (in practice, 100) when the robot reaches the final goal state. At all other intermediate states, the extrinsic rewards are set to 0. Formally, $r_t^e(s_{t-1}, a_{t-1}, s_t, g) = 100$ if and only if $s_t$ is a goal state, otherwise $r_t^e(s_{t-1}, a_{t-1}, s_t, g) = 0$.

**Intrinsic rewards $r^i$.** To facilitate the robot perform the sub-task, i.e. approaching the object specified in the sub-goal $sg$ which shows in the robot's current view, we adopt the following reward function (Ye et al., 2018a) as the intrinsic rewards. To be specific, let $ar_t$ be the area of the object in robot's view at time step $t$ (known as $s_t$), the intrinsic reward $r_t^i(s_{t-1}, a_{t-1}, s_t, sg) = ar_t$ if and only if $ar_t > ar_{t-1}, ar_{t-2}, ...ar_0$, otherwise $r_t^i(s_{t-1}, a_{t-1}, s_t, sg) = 0$. Note that $ar_t$ can be easily calculated based on the robot's detection outputs, e.g. the size of the detected bounding box. In practice, we normalize $ar_t$ to the range of $[0, 100]$.

## 2.3 Model Formulation

We formulate the ROS task in terms of the two rewards introduced in Sec. 2.2. When the robot starts from an initial state $s_0$, it proposes a sub-goal $sg_0$ aiming to achieve the final goal $g$ (locating and approaching the target object). To achieve the final goal, we can optimize the discounted cumulative extrinsic rewards, expected over all trajectories starting at state $s_0$ and sub-goal $sg_0$, which is $\mathbb{E}[\sum_{t=0}^{\infty} \gamma^t r_{t+1}^e | s_0, g, sg_0]$. If and only if the robot takes minimal steps to the goal state, the discounted cumulative extrinsic rewards are thus maximized.

The discounted cumulative extrinsic rewards is also known as the state action value $Q_h^e$ (Sutton & Barto, 2018) for our high-level layer, i.e. $\mathbb{E}[\sum_{t=0}^{\infty} \gamma^t r_{t+1}^e | s_0 = s, g = g, sg_0 = sg] = Q_h^e(s, g, sg)$. Following the option-critic framework (Bacon et al., 2017), we unroll the $Q_h^e(s, g, sg)$ as,

$$
\begin{aligned}
Q_h^e(s, g, sg) &= \sum_a \pi_l(a|s, g, sg) \mathbb{E}[\sum_{t=0}^{\infty} \gamma^t r_{t+1}^e | s_0 = s, g = g, sg_0 = sg, a_0 = a] \\
&= \sum_a \pi_l(a|s, g, sg) Q_l^e(s, g, sg, a),
\end{aligned}
\tag{1}
$$

where the state action value $Q_l^e(s, g, sg, a)$ for our low-level layer is the discounted cumulative extrinsic rewards after taking action $a$ under the state $s$, goal $g$ and sub-goal $sg$. Given the transition probability $P(s'|s, a)$ which denotes the probability of being state $s'$ after taking action $a$ at state $s$, $Q_l^e(s, g, sg, a)$ can be further formulated as,

$$
Q_l^e(s, g, sg, a) = \sum_{s'} P(s'|s, a)[r^e(s, a, s', g) + \gamma U(g, sg, s')],
$$

$$
U(g, sg, s') = (1 - term(s', g, sg)) Q_h^e(s', g, sg) +
$$
$$
term(s', g, sg) V_h^e(s', g),
$$

$$
V_h^e(s', g) = \sum_{sg'} \pi_h(sg'|s', g) Q_h^e(s', g, sg').
\tag{2}
$$

We parameterize $\pi_l(a|s, g, sg)$ and $term(s, g, sg)$ as $\theta_{\pi_l}$ and $\theta_t$ respectively. According to Bacon et al. (2017), $\theta_{\pi_l}$ and $\theta_t$ can be updated by Equation 3 and 4 to optimize $Q_h^e(s, g, sg)$. We refer interested readers to Bacon et al. (2017) for more details.

$$
\theta_{\pi_l} \leftarrow \theta_{\pi_l} + \nabla_{\theta_{\pi_l}} \log \pi_{\theta_{\pi_l}}(a|s, g, sg)(Q_l^e(s, g, sg, a) - Q_h^e(s, g, sg)).
\tag{3}
$$

$$
\theta_t \leftarrow \theta_t - \nabla_{\theta_t} term_{\theta_t}(s', g, sg)(Q_h^e(s', g, sg) - V_h^e(s', g)).
\tag{4}
$$

In addition, we adopt Q-learning method (Mnih et al., 2015) to learn $Q_h^e(s, g, sg)$ and generate sub-goal $sg = \arg\max_{sg} Q_h^e(s, g, sg)$. We parameterize $Q_h^e(s, g, sg)$ with $\theta_{he}$ and update its value towards the 1-step extrinsic return $R_1^e = r^e(s, a, s', g) + \gamma U_{\theta_{he}}(g, sg, s')$, and consequently $\theta_{he}$ can be updated as follows.

$$
\theta_{he} \leftarrow \theta_{he} - \nabla_{\theta_{he}}[R_1^e - Q_{\theta_{he}}(s, g, sg)]^2.
\tag{5}
$$

For the low-level policy layer, when we build the hierarchical policy from the sub-goal space, we inherently assume that achieving the sub-goal is helpful for achieving the final goal. We define the sub-goal as approaching a related object by doing so the robot is more likely to observe the target one. Without loss of generality, we introduce $\alpha \in [0, 1]$ as the proportion of how much effort the low-level layer puts in achieving the sub-goal. Meanwhile, the low-level layer keeps $(1 - \alpha)$ of its effort for achieving the final goal, which yields a $\alpha$ mixed overall loss.

Similarly, to learn the desired policy for the low-level layer, we optimize the discounted cumulative hybrid rewards, $\mathbb{E}[\sum_{t=0}^{\infty} \gamma^t((1-\alpha)r_{t+1}^e + \alpha r_{t+1}^i)|s_0, g, sg_0, a_0]$. We denote it as $Q_l(s, g, sg, a)$, and further represent it by $Q_l^e(s, g, sg, a)$ and $Q_l^i(s, g, sg, a)$ as in Equation 6.

$$
\begin{aligned}
Q_l(s, g, sg, a) =& \mathbb{E}[\sum_{t=0}^{\infty} \gamma^t((1-\alpha)r_{t+1}^e + \alpha r_{t+1}^i)|s_0 = s, g = g, sg_0 = sg, a_0 = a] \\
=& (1-\alpha)\mathbb{E}[\sum_{t=0}^{\infty} \gamma^t r_{t+1}^e|s_0 = s, g = g, sg_0 = sg, a_0 = a] + \\
& \alpha\mathbb{E}[\sum_{t=0}^{\infty} \gamma^t r_{t+1}^i|s_0 = s, g = g, sg_0 = sg, a_0 = a] \\
=& (1-\alpha)Q_l^e(s, g, sg, a) + \alpha Q_l^i(s, g, sg, a).
\end{aligned}
\tag{6}
$$

As a result, to optimize $Q_l(s, g, sg, a)$, we can optimize $Q_l^e(s, g, sg, a)$ and $Q_l^i(s, g, sg, a)$ one by one. With the policy gradient method, $\theta_{\pi_l}$ can be updated using Equation 7 and Equation 8 respectively (Mnih et al., 2016).

$$
\theta_{\pi_l} \leftarrow \theta_{\pi_l} + \nabla_{\theta_{\pi_l}} \log \pi_{\theta_{\pi_l}}(a|s, g, sg)(Q_l^e(s, g, sg, a) - V_l^e(s, g, sg)),
\tag{7}
$$

$$
\theta_{\pi_l} \leftarrow \theta_{\pi_l} + \nabla_{\theta_{\pi_l}} \log \pi_{\theta_{\pi_l}}(a|s, g, ag)(Q_l^i(s, g, sg, a) - V_l^i(s, g, sg)).
\tag{8}
$$

Note that $V_l^e(s, g, sg) = \sum_a \pi_l(a|s, g, sg)Q_l^e(s, g, sg, a)$ by definition, therefore $V_l^e(s, g, sg) = Q_h^e(s, g, sg)$ according to Equation 1. Thus, the update of $\theta_{\pi_l}$ in Equation 7 for optimizing $Q_l^e(s, g, sg, a)$ is exactly the same as that in Equation 3 for optimizing $Q_h^e(s, g, sg)$. At last, we combine the two updates of $\theta_{\pi_l}$ according to Equation 6. Equation 9 gives the final update.

$$
\begin{aligned}
\theta_{\pi_l} \leftarrow \theta_{\pi_l} + \nabla_{\theta_{\pi_l}} \log \pi_{\theta_{\pi_l}}(a|s, g, sg)* \\
[(1-\alpha)(Q_l^e(s, g, sg, a) - Q_h^e(s, g, sg)) + \\
\alpha(Q_l^i(s, g, sg, a) - V_l^i(s, g, sg))].
\end{aligned}
\tag{9}
$$

To conclude, we adopt Equation 4, 5 and 9 to update $\theta_t$, $\theta_{he}$ and $\theta_{\pi_l}$ for learning $term(s, g, sg)$, $Q_h^e(s, g, sg)$ and $\pi_l(a|s, g, sg)$ respectively.

## 2.4 Network Architecture

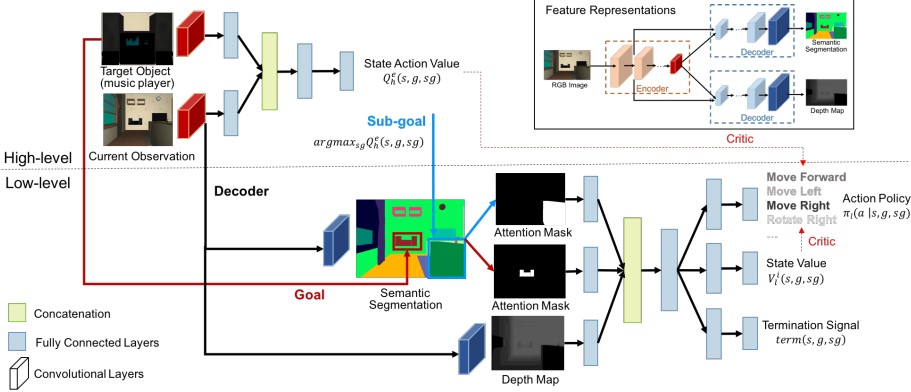

Figure 2: Network architecture of our hierarchical reinforcement learning model.

Since the image captured by the robot's camera serves as the robot's current state, we adopt deep neural networks to handle this high dimensional inputs and to approximate $term(s, g, sg)$, $Q_h^e(s, g, sg)$ and $\pi_l(a|s, g, sg)$. To update the model parameters according to Equation 4, 5 and 9, we still need the value of $V_h^e(s, g)$, $Q_l^e(s, g, sg, a)$, $Q_l^i(s, g, sg, a)$ and $V_l^i(s, g, sg)$. Here, we use deep neural networks to approximate $V_l^i(s, g, sg)$, and estimate the remaining (Equation 10).

$$
\begin{aligned}
V_h^e(s, g) =& \max_{sg} Q_h^e(s, g, sg), \\
Q_l^e(s, g, sg, a) =& r^e(s, a, s', g) + \gamma U(g, sg, s'), \\
Q_l^i(s, g, sg, a) =& r^i(s, a, s', sg) + \gamma V_l^i(s, g, sg).
\end{aligned}
\tag{10}
$$

Fig. 2 illustrates our network architecture. We first extract multiple feature representations from the encoder-decoder network (shown in Fig. 2 upper right corner) for the image inputs. Our high-level network takes the outputs from the encoder as the feature representations of the robot's current view and the target object image accordingly. The two feature representations are then fed into two different fully connected layers respectively and the outputs are then concatenated into a joint vector before attaching another fully connected layer to generate an embedding fusion. We feed the embedding fusion into one additional fully connected layer to approximate $Q_h^e(s, g, sg)$. For low-level model, we take the outputs of the two decoders (a semantic segmentation decoder and a depth prediction decoder) as the feature representations of the robot's current view. Based on the goal (i.e. the target object) and the sub-goal proposed by the high-level network, we generate two attention masks using the semantic segmentation module. Note that if the sub-goal is proposed as the candidate non-goal-driven exploration, there is no sub-goal attention in hope that the robot can learn to explore. We take the two attention masks, as well as the depth map as the inputs to our low-level network, which shares a similar architecture as our high-level network except that the low-level network has three branches attached after the embedding fusion for approximating $\pi_l(a|s, g, sg)$, $V_l^i(s, g, sg)$ and $term(s, g, sg)$ respectively. Each branch consists of two fully connected layers.

We follow Equation 4, 5 and 9 to learn $term(s, g, sg)$, $Q_h^e(s, g, sg)$ and $\pi_l(a|s, g, sg)$. In addition, to learn $V_l^i(s, g, sg)$, we update the estimated value towards the 1-step intrinsic return $R_1^i = r^i(s, a, s', sg) + \gamma V_{\theta_{li}}(s', g, sg)$. As a result, the parameter $\theta_{li}$ could then be updated as,

$$\theta_{li} \leftarrow \theta_{li} - \nabla_{\theta_{li}}[R_1^i - V_{\theta_{li}}(s, g, sg)]^2. \tag{11}$$

## 3 EXPERIMENTS

### 3.1 DATASET

We validate our framework on the simulation platform House3D (Wu et al., 2018). House3D consists of rich indoor environments with diverse layouts for a virtual robot to navigate. In each indoor environment, a variety of objects are scattered at many locations, such as *television, sofa, desk*. While navigating, the robot has a first-person view RGB image as its observation. The simulator also provides the robot with the ground truth semantic segmentation and depth map corresponding to the RGB image. The RGB images, as well as the semantic segmentation and depth maps can be used as the training data to learn the encoder-decoder network (shown in Fig. 2 upper right corner) for the feature representation extraction as we mentioned in Sec. 2.4. In addition, the trained model, specifically the semantic segmentation prediction, can be used as the robot's detection system.

To validate our proposed method in learning hierarchical policy for object search, we conduct the experiments in indoor environments where the objects' placements are in accordance with the real-world scenario. For example, the *television* is placed close to the *sofa*, and is likely occluded by the *sofa* at many viewpoints. In such a way, to search the target object *television*, the robot could approach *sofa* first to increase the likelihood of seeing the *television*.

We consider discrete actions for the robot to navigate in this environment. Specifically, the robot moves forward / backward / left / right 0.2 meters, or rotates 90 degrees every time. It also discretizes each environment into a certain number of reachable locations, as shown in Fig. 3.

### 3.2 EXPERIMENTAL SETTING

We compare the following methods and variants:

**RANDOM.** At each time step, the robot ignores its observation and performs a random action.

**HIGH-LEVEL ONLY.** We remove our low-level network and train the high-level network to predict the state action value $Q^e(s, g, a)$ with the DQN method (Mnih et al., 2015).

**LOW-LEVEL ONLY.** Only the low-level network is trained to generate an action policy for object search without hierarchy. Since there is no sub-goal provided to the low-level network, we further block the sub-goal input channel. A recent work from Ye et al. (2018a) shows that it serves as a stronger baseline than Zhu et al. (2017).

**OPTION-CRITIC** (Bacon et al., 2017). This method aims to build the hierarchy automatically. It learns the desired policy by maximizing the discounted cumulative extrinsic rewards, while no intrinsic rewards are involved. When the high-level network proposes a sub-goal (option), the corresponding option policy is learned by maximizing the extrinsic rewards and acting towards the goal. The option policy stops according to the predicted termination signal. Here, the sub-goal doesn't have much interpretability, it can be seen as a temporal abstraction.

**HRL** (Kulkarni et al., 2016). Different from OPTION-CRITIC, this method works on both extrinsic rewards and intrinsic rewards. The high-level network maximizes the discounted cumulative extrinsic rewards to reach the goal, while the low-level network maximizes the discounted cumulative intrinsic rewards to achieve the sub-goal that is proposed by the high-level network. The low-level network stops either when the sub-goal is achieved, or a predefined maximum number of steps have been performed.

**HRL WITH STOP** (Das et al., 2018b). This method is similar to HRL, while the low-level network has an additional action *stop* in its action space. When the low-level network produces the *stop* action, it stops and returns the control to the high-level network. Otherwise, the low-level network continues until it has performed a predefined maximum number of steps. Note that in Das et al. (2018b), the authors adopted an imitation learning method to warm-start the hierarchical policy learning. For fair comparison, we don't provide any imitation signals in our experiments, and we learn the policy purely through the reinforcement learning method.

**OUR METHOD** follows Sec 2. To identify the role of the $\alpha$ value, we compare multiple variants of our method, namely setting $\alpha = 0, 0.2, 0.4, 0.6, 0.8, 1$ respectively, where the higher $\alpha$ value denotes the more effort the low-level network devotes to achieving the sub-goal. It is also worth to mention that when setting $\alpha = 0$, our method shares a similar idea with OPTION-CRITIC. However, our low-level network still learns to approximate the state value $V_l^i(s, g, sg)$ which is supervised by the intrinsic rewards. When setting $\alpha = 1$, our method is similar to HRL. Yet our low-level network learns a termination signal $term(s, g, sg)$, which is supervised by the extrinsic rewards.

Experiments are conducted under two settings:

**SETTING-STSE** (Singe Target in Single Environment). To compare the efficacy in learning the object search policy, we train each method to learn a policy for searching a specific target object. Without loss of generality, we randomly choose 1 target object from a specific environment. We evaluate each method in terms of the network for searching the specific target object.

**SETTING-MTME**(Multiple Targets in Multiple Environments). To further evaluate, we train every method under different environments to search different target objects. To be specific, we randomly choose 24 target objects in 4 different environments (6 each) as our training data. We evaluate each method in terms of the network with the 24 different environment-target object pairs being taken as the inputs randomly.

For each experiment, we periodically increase the distance between the target object and the robot's starting position during training time which is also known as the curriculum learning paradigm. During testing time, we randomly choose 100 starting positions. We set the maximum number of all atomic action steps to 1000. For method HRL and HRL WITH STOP, we set the maximum number of each low-level steps to 50 to ensure the proposed sub-goals can be achieved. The robot stops either it reaches the goal state (success case) or it runs out of the maximum number of all atomic action steps (failure case). We implement our algorithm using Tensorflow toolbox and conduct all the experiments with Nvidia V100 GPUs and 16 Intel Xeon E5-2680 v4 CPU cores.

### 3.3 EXPERIMENTAL RESULTS AND DISCUSSION

Since we formulate the object search problem as maximizing the discounted cumulative extrinsic rewards, we take the Average discounted cumulative extrinsic Rewards (AR) as one of the evaluation metrics, calculated by:

$$\frac{1}{N} \sum_{i=1}^{N} \sum_{t=0}^{\infty} \gamma^t r_{t+1}^e = \frac{1}{N} \sum_{i=1}^{N} \mathbb{1}(success) \gamma^{\#steps} * 100, \tag{12}$$

where $\gamma \in (0, 1]$ is the discount factor. From the perspective of the evaluation metric, it can also be seen as a trade-off between the success rate metric and the average steps metric. With the higher

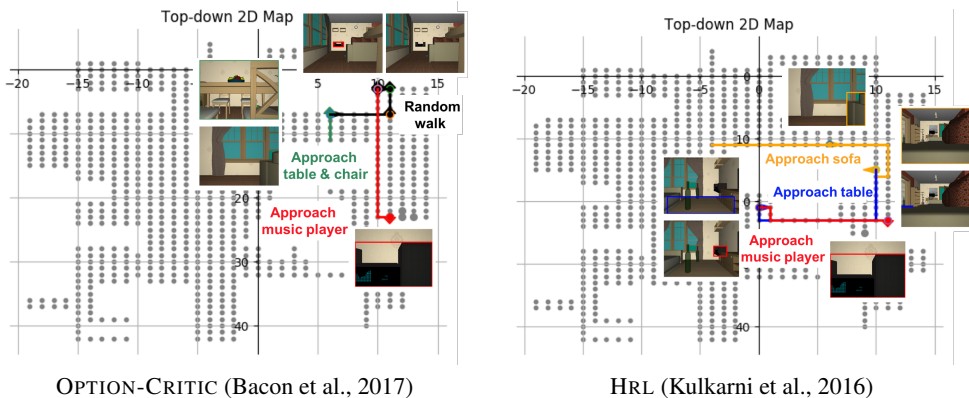

OPTION-CRITIC (Bacon et al., 2017)          HRL (Kulkarni et al., 2016)

Figure 3: Trajectories generated from OPTION-CRITIC and HRL for searching the target object *music player* from the same starting position under the SETTING-STSE.

value of $\gamma$, the average steps metric weighs more, and vice versa. In our experiments, we set $\gamma = 0.99$.

In addition, we also report the following widely used evaluation metrics. Success Rate (SR). Average Steps over all successful cases (AS). Success weighted by inverse Path Length (SPL) (Anderson et al., 2018a), which is calculated as $\frac{1}{N} \sum_{i=1}^{N} S_i \frac{l_i}{max(l_i, p_i)}$. Here, $S_i$ is the binary indicator of success in episode $i$, $l_i$ and $p_i$ are the lengths of the shortest path and the path actually taken by the robot. We adopt the number of the action steps as the path length. As a result, SPL also trades-off success rate against average steps.

Fig. 4 reports the training performance (i.e. AR and SPL) of our method with different $\alpha$ values under SETTING-STSE. From Fig. 4, we observe that the model with $\alpha = 0.8$ achieves much better performance, indicating that a proper trade-off between achieving the goal and the sub-goal is needed for our hierarchical method to perform the object search well. When $\alpha = 0$, the model performs worst. It is because that the low-level layer needs to choose the actions that are suited for achieving the goal, and as a result the learning process still suffers from the sparse extrinsic rewards setting. When $\alpha = 1$, the model performs much better at the earlier training stage where the target object is closer to robot's starting position. We explain it as when $\alpha = 1$, the navigation actions generated by the low-level network are fully driven by the intrinsic rewards in order to achieve the sub-goals. When the target object is nearby, the proposed sub-goals are more likely to be consistent with the final goal and thus achieving the sub-goals is helpful for achieving the goal. While the target object is far away, the proposed sub-goals are much more noisy. Then focus only on achieving the sub-goals without keeping the final goal in mind may hurt the performance.

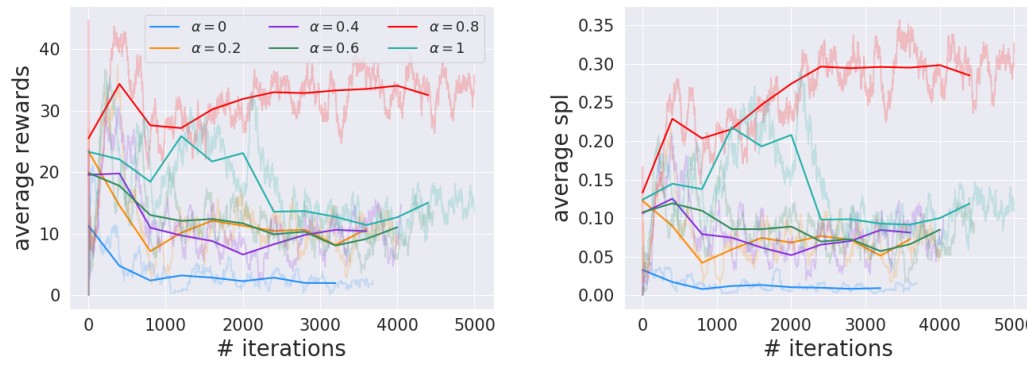

Figure 4: The average discounted cumulative extrinsic rewards (left) and average SPL (right) of our method with different $\alpha$ values under SETTING-STSE.

Table 1: The performance of all methods under SETTING-STSE and SETTING-MTME.

| Method | SETTING-STSE | | | | SETTING-MTME | | | |
|---|---|---|---|---|---|---|---|---|
| | SR↑ | AS↓ | SPL↑ | AR↑ | SR↑ | AS↓ | SPL↑ | AR↑ |
| ORACLE | 1.00 | 26.74 | 1.00 | 77.72 | 1.00 | 22.42 | 1.00 | 81.39 |
| RANDOM | 0.16 | 357.56 | 0.01 | 3.91 | 0.23 | 372.65 | 0.01 | 4.07 |
| HIGH-LEVEL ONLY | 0.08 | **71.25** | 0.04 | 5.51 | 0.21 | **202.71** | 0.03 | 8.18 |
| LOW-LEVEL ONLY | 0.16 | 315.75 | 0.03 | 6.57 | 0.25 | 355.40 | 0.01 | 4.93 |
| OPTION-CRITIC | 0.25 | 388.28 | 0.02 | 4.74 | 0.28 | 429.21 | 0.01 | 4.37 |
| HRL | 0.58 | 359.52 | 0.08 | 11.03 | 0.35 | 340.34 | 0.05 | 10.00 |
| HRL WITH STOP | 0.66 | 338.33 | 0.13 | 18.02 | **0.36** | 370.37 | 0.04 | 5.80 |
| **Ours** | | | | | | | | |
| $\alpha = 0.8$ | **0.77** | 289.56 | **0.20** | **23.34** | 0.34 | 208.32 | **0.07** | **12.96** |

We set $\alpha = 0.8$ in the following experiments. Table 1 shows the comparison of all methods under both settings. Table 1 (left) shows that under SETTING-STSE, our method ($\alpha = 0.8$) outperforms all previous methods in terms of SR, SPL, AR, and fairly in terms of AS since HIGHLEVEL ONLY method doesn't perform well in general as its low SR, SPL and AR suggest. Moreover, we also observe that OPTION-CRITIC doesn't perform well as other hierarchical methods (HRL, HRL WITH STOP and our method), indicating that the prior knowledge provided by human is vital in guiding the robot to perform the challenge tasks. Compared to HRL WITH STOP, HRL doesn't stop the robot at a more valuable state timely and for which the performance is worse. However, simply add a *stop* action in the atomic action space is not much efficient when compared with our method. Under the more challenge SETTING-MTME, though our method has a slightly lower SR and higher AS from Table 1 (right), our method achieves higher SPL and AR that both trade-off the SR against AS, which also fairly demonstrates the superiority of our method.

We further depict sample qualitative results in Fig. 1 (right) and Fig. 3, which show that our method yields a more concise trajectory compare to other methods for the ROS task.

## 4 RELATED WORK

Our work is closely related to two major research thrusts: hierarchical reinforcement learning and target-driven visual navigation.

**Hierarchical reinforcement learning.** Previous work has studied hierarchical reinforcement learning in many different ways. One is to come up with efficient methods to accelerate the learning process of the general hierarchical reinforcement learning scheme. As in Nachum et al. (2018b), the authors introduced an off-policy correction method. Levy et al. (2017) and Levy et al. (2018) proposed to use Hindsight Experience Replay to facilitate learning at multiple time scales. Though these methods' performance are impressive, they typically assume the sub-goal space for the higher level policy is a subspace of the state space. However, in the ROS task, the RL system takes the image as the state representation, these methods are not directly applicable since the higher layer can hardly propose an image as a sub-goal for the lower layer to achieve.

Other methods designate a separate sub-goal space for hierarchical reinforcement learning. For example, Kulkarni et al. (2016) defined the sub-goal space in the space of entities and relations, such as the "reach" relation they used for their Atari game experiment. Sub-tasks and their relations were provided as inputs in Andreas et al. (2017) and Sohn et al. (2018). Closer related to our work, Das et al. (2018b) adopted {*exit-room, find-room, find-object, answer*} as the sub-goal space to learn a hierarchical policy for the Embodied Question Answering task. For the same task, Gordon et al. (2018) chose {*navigate, scan, detect, manipulate, answer*} as the possible sub-tasks, while the reinforcement learning methods were mainly applied for learning high-level policy. More recently, Gordon et al. (2019) integrated symbolic planning as an additional sub-task.

On the other side, attempts have been made to learn the sub-tasks automatically. These sub-tasks are referred to as temporal abstractions. Bacon et al. (2017) proposed the option-critic framework to autonomously discover the specified number of temporal abstractions. Osa et al. (2019) learned

the temporal abstractions through advantage-weighted information maximization. Nachum et al. (2018a) addressed the sub-goal representation learning problem. With the learned representation, their hierarchical policies are shown to approach the optimal performance within a bounded error.

Motivated by these works, we fully utilize the human specified sub-goal space to make the hierarchy better interpretable. Meanwhile, to avoid being misled by the possibly imperfect human knowledge, we also leverage the benefits from the automatic temporal abstraction learning methods, which yields a hybrid one.

**Target-driven visual navigation.** Deep reinforcement learning has been studied extensively for the target-driven visual navigation task. Mirowski et al. (2016) proposed learning depth prediction and action policy jointly for visual navigation. Zhu et al. (2017) introduced a target-driven deep reinforcement learning framework to enable a robot to reach an image-specified location. The Embodied Question Answering or Interactive Question Answering task is introduced and addressed in Das et al. (2018a),Das et al. (2018b),Gordon et al. (2018), Gordon et al. (2019), Yu et al. (2019) and Wijmans et al. (2019), where the robot needs to navigate an environment to answer an unconstrained natural language question. Gupta et al. (2017), Ye et al. (2018b), Mousavian et al. (2018) and Ye et al. (2018a) focused on the robotic object search and/or object approaching task. Anderson et al. (2018b),Wang et al. (2018b) and Wang et al. (2018a) focused on natural language guided visual navigation. Among all of these cited work, most of them plan over the atomic actions for navigation (Mirowski et al., 2016; Zhu et al., 2017; Das et al., 2018a; Wijmans et al., 2019; Gupta et al., 2017; Ye et al., 2018b; Mousavian et al., 2018; Ye et al., 2018a; Anderson et al., 2018b; Wang et al., 2018a) without hierarchical modeling, and thus lack interpretability. While Gordon et al. (2018) and Gordon et al. (2019) studied hierarchical policies, they manually constructed the hierarchy with high-level and low-level controllers, and a standard reinforcement learning method is applied on high-level layer to decide which low-level controller to invoke. Das et al. (2018b) also studied hierarchical reinforcement learning to generate explainable hierarchical policies. Along with these mentioned methods that build upon human specified sub-goal space, we further consider the likelihood that these manual sub-goals may not lead to the optimal steps towards the final goal, and we model it explicitly in our framework.

More importantly, many of the previous works assume that the agent can access the full information of the environments during the training time, either by defining the reward function with the distance between the agent's current location and the target location (Mousavian et al., 2018; Wang et al., 2018b;a), and/or adopting shortest path as the supervised signal for pre-training (Das et al., 2018a;b; Gordon et al., 2018; Anderson et al., 2018b; Wang et al., 2018a; Yu et al., 2019). Nevertheless, for applications in real-world environments, collecting all the information is unarguably expensive and sometimes impossible. *We would like to stress upon the point that our model does not assume any environment information accessible even during the training process, which levels up the ROS' difficulty.*

## 5 CONCLUSION AND FUTURE WORK

In this paper, we present a novel two-layer hierarchical policy learning framework that builds on intrinsic and extrinsic rewards. The framework fully utilizes the prior knowledge provided by human to make the hierarchy scheme interpretable. When the high-level layer plans over the human specified sub-goal space to achieve the goal, the low-level layer plans over the atomic actions by taking both the sub-goal proposed by the high-level layer and the goal into the consideration. The framework is general and we validate it in the context of the object search task. The empirical experiments on House3D platform demonstrate the efficacy and efficiency of our proposed framework.

The current work also opens several avenues for future study. First, while we adopt a fixed $\alpha$ value in our experiments to indicate the importance of the sub-goals compared to the goal, the $\alpha$ value can be adaptively updated while approaching different sub-goals or even be learned from common-sense knowledge. We intend to extend our work towards integrating top-down human knowledge together with the human specified sub-goal space to facilitate the object search task with higher efficiency.

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

# A    NETWORK ARCHITECTURE AND TRAINING DETAILS

Figure 5: Network architecture of our hierarchical reinforcement learning model.

**Feature representation network.** The network is shown in Fig. 5 upper right corner. It is trained to predict the semantic segmentation and depth map from a single RGB image. The network builds upon Chen et al. (2018) with an additional decoder branch to output the depth map. The two decoder branches share the the same architecture except that one decoder branch pipelines to a classification output layer for semantic segmentation and the other adopts a regression output layer for depth prediction. We adopt the cross entropy loss function for semantic segmentation and mean-square error loss function for depth prediction and train the network with the images sampled from $100$ environments from House3D (Wu et al., 2018).The trained network is then deployed for extracting the feature representations and the semantic segmentation prediction is further adopted as the robot's detection system.

**High-level network.** The network is shown in Fig. 5 upper part. The encoder of the feature representation network outputs a $256$ dimensional feature for either target object image or the robot's current observation image. For each input stream, we concatenate the features of $4$ history frames and then project the $1024$ dimensional vector down to a $512$ dimensional vector. The two $512$ dimensional vectors from both input streams are then concatenated and aggregated into a singe $512$ dimensional joint vector. The joint vector is further projected into $78$ state action values. ($78$ is the number of the objects that being of our interest.)

**Low-level network.** The network is shown in Fig. 5 lower part. We extract the two attention masks from the predicted semantic segmentation with the semantic label of the target object and the object specified in the sub-goal. The two attention masks and the predicted depth map are then resized to $10$ by $10$. For each input stream, we concatenate the features of $4$ history frames and then project the $400$ dimensional vector down to a $512$ dimensional vector. The three $512$ dimensional vectors from three input streams are then concatenated and aggregated into a single $512$ dimensional joint vector. The joint vector is further passed through three branches. For all three branches, we first project the $512$ dimensional joint vector into a $20$ dimensional vector. In the first branch, we further project the $20$ dimensional vector into $6$ atomic action policy outputs, i.e. probability over atomic actions. In the second branch, we project the $20$ dimensional vector into a single state value output. In the third branch, we project the $20$ dimensional vector into a single signal output as termination, i.e. probability to terminate.

We train this hierarchical network with a RMSProp optimizer of learning rate $1 \times 10^{-4}$, and we reduce the learning rate by an order of magnitude every $1000$ episodes.

# B    QUALITATIVE RESULTS

Fig. 6 shows some qualitative results of our method. The robot can only access the first-person view RGB images. We show the top-down 2D map for better visualization. From Fig. 6, we can see that

the robot chooses to approach a related object first before detecting the target object. Once the target object is detected, the robot then opts to approach the target object immediately.

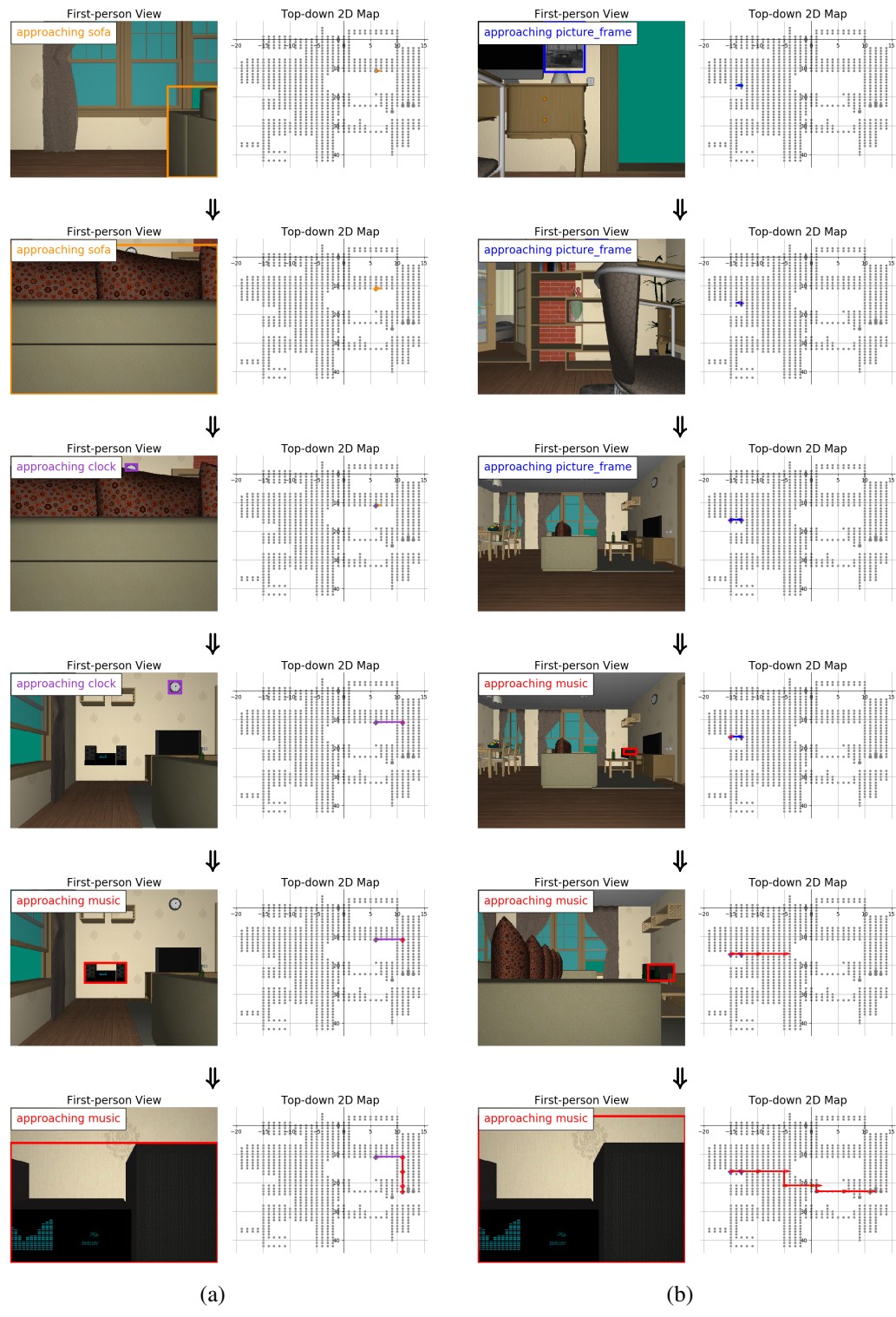

(a)                                                      (b)

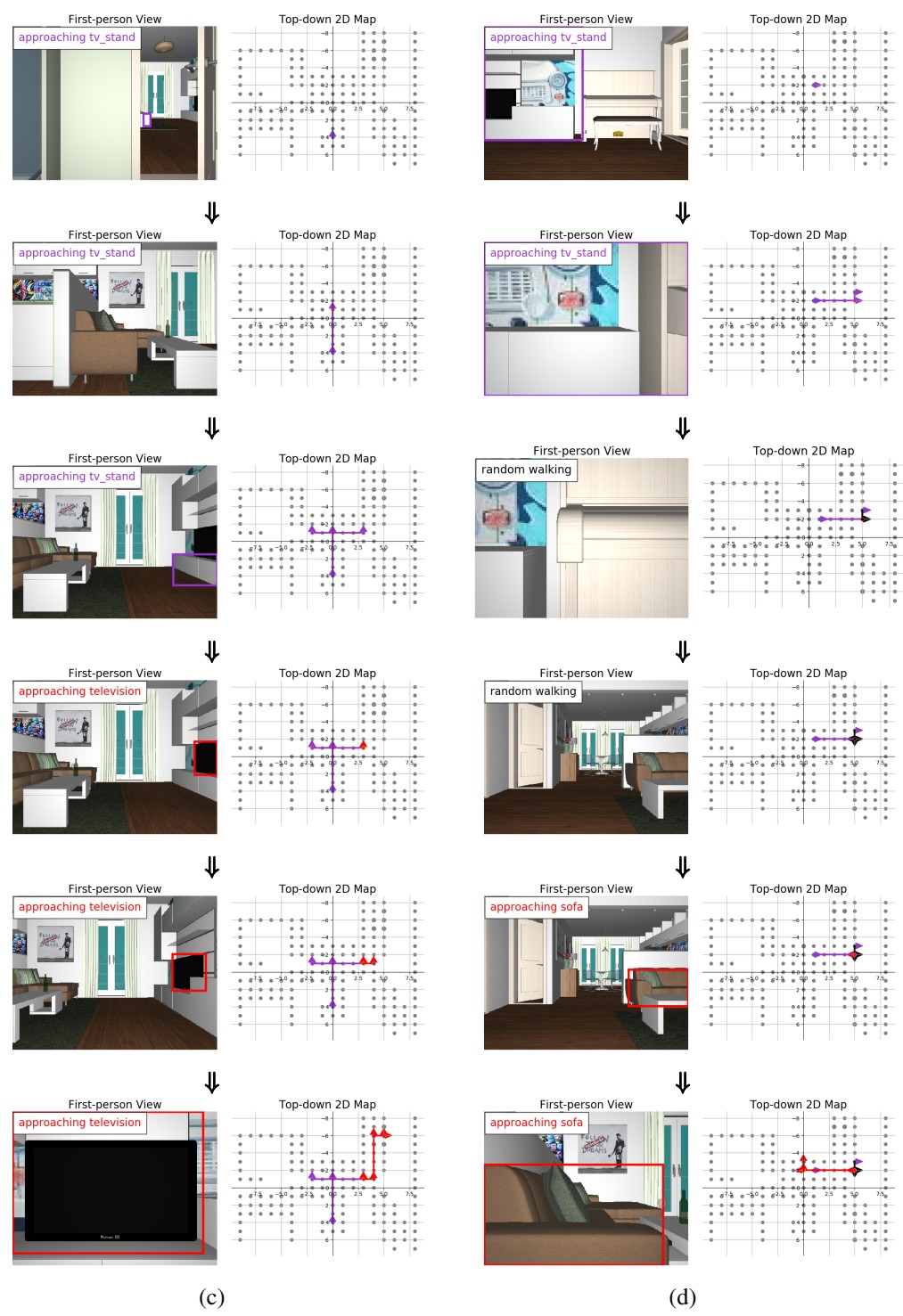

Figure 6: Trajectories generated by our method for the robot to search (a) *music* (b) *music* (c) *television* and (d) *sofa*.

