# OpenReview forum: "ROS-HPL: Robotic Object Search with Hierarchical Policy Learning and Intrinsic-Extrinsic Modeling"
_ICLR.cc/2020/Conference — Reject_

### Official Review · AnonReviewer3 · 2019-10-23
**Official Blind Review #3**

**Rating:** 3

**Review:**

This paper proposed a hierarchical approach to perform robotic object search (ROS).
The idea is to use a high-level policy which produces subgoals and a low-level policy which produces atomic actions conditioned on both the subgoals and the true goal, and which is trained with a weighted sum of the original extrinsic reward and a reward for reaching the subgoal. Subgoals are consist of different objects in the field of view which the robot can choose to approach.
The approach is evaluated on the House3D dataset, where it is shown to perform well.

Recommendation: weak reject.

This isn't a bad paper, but I'm not sure it will be of broad interest at ICLR.
It is very specific to ROS problem and House3D dataset and doesn't seem to propose a general algorithm which can be broadly applicable elsewhere. The mechanism for generating subgoals and training the low-level policy is very task dependent (subgoals are constrained to be objects in the field of view, the intrinsic reward for training the low level policy is dependent on the size of the bounding box of the object defining the subgoal). While this probably accounts for improved performance on the House3D dataset, I think the audience of ICLR would be more interested in a general approach which can be used in many different domains (even at the cost of performing less well on a specific domain than something more tailored). This paper may be a better fit for a robotics conference.

Another point concerning the experimental evaluation. Sparsity of the rewards is mentioned as a main motivation for the hierarchical approach. However, there are a number of methods which use exploration bonuses to address this issue (pseudocounts, random network distillation, ICM etc. [1, 2, 3]). At least one of these should be included as a baseline.

Specific comments:
- using two letters for denote a single variable is confusing, since it seems like a product. I.e. using "sg" to denote a subgoal, "at_t" to denote area. Please use a single letter and add subscripts if necessary to disambiguate.
- in the various equations, please use "\log" instead of "log" so that it is not italicized.
- bottom of page 4: "Q-leaning" -> "Q-learning"
- page 2: "way pf" -> "way of"
- please use more informative names for Settings A/B
- First paragraph in Section 2: "hierarchical policy for the robot to perform the object search, motivated by how human beings conduct object search". Saying the method is similar to how humans behave is a fairly big claim that should be substantiated by appropriate references, or not made at all.


References:
[1] https://arxiv.org/abs/1810.12894
[2] https://arxiv.org/abs/1703.01310
[3] https://arxiv.org/abs/1705.05363


**Experience Assessment:**

I do not know much about this area.

**Review Assessment: Checking Correctness Of Derivations And Theory:**

N/A

**Review Assessment: Checking Correctness Of Experiments:**

I assessed the sensibility of the experiments.

**Review Assessment: Thoroughness In Paper Reading:**

I read the paper at least twice and used my best judgement in assessing the paper.

---

> ### Author Response · Authors · 2019-11-14
> **Response to Reviewer #3**
>
> Thank you for the constructive comments. We address the concerns as below,
> 1. We would like to reiterate the contribution of our work is the general hierarchical RL framework we proposed that 1) builds upon the human specified sub-goal space to incorporate human prior knowledge, that makes such a challenge task easier to learn and the solution interpretable; 2) optimizes with the intrinsic and the extrinsic rewards jointly to overcome the lingering inconsistency coming from the unsatisfied prior knowledge.
>     The motivation is that we observed for challenge tasks, previous work that learn to build the hierarchy and the sub-goal space themselves without prior knowledge achieve poor performance, while other work that fully adopt the human-specified sub-goal space ignore the possibility that the prior knowledge may not be accurate enough for learning the hierarchical policy.
>     Robotic object search is such a task where humans can specify the sub-goal space as approaching a currently visible object based on the prior knowledge that approaching a related object may increase the probability of seeing the target object. However, such prior knowledge and sub-goal space is not satisfied for learning an optimal hierarchical policy, i.e. to approach the target object, the robot does not necessarily to be close enough to a sub-goal specified object. We therefore demonstrate our proposed hierarchical RL framework on this task, and we believe  the more challenging the task is, the more difficult it is for humans to provide satisfied prior knowledge and designate sub-goal space.
>     Overall, we believe our proposed hierarchical RL framework shed light on learning challenge tasks by utilizing human prior knowledge in a proper way. The definitions of  the sub-goals or reward functions can be modified willingly in accordance with the different tasks.
>
> 2. The focus of this paper is to learn a hierarchical policy for robotic object search task that is able to address the sparse reward issue, so we compared our method with other hierarchical RL methods rather than all methods addressing the sparse reward issue.
>
> 3. Thanks again for pointing out the typos, and we have updated our paper properly.

---

### Official Review · AnonReviewer2 · 2019-10-28
**Official Blind Review #2**

**Rating:** 3

**Review:**

Summary:
The paper proposes an intuitive 2-layer hierarchy for robotic object search. The high-level policy does subgoal selection, whereas the low-level layer handles explicit control. Notably, the low-level policy is trained to be aware of both the subgoal and the final goal. The authors conducted a series of ablations, demonstrating the value of training the low-level policy to be final goal-aware, and empirically demonstrated the strength of their method compared to other baselines.

Decision: Weak Reject. The idea is intuitive and seems to be empirically successful (on some metrics). My primary concern is that the work appears incremental when compared to the baselines HRL and HRL with Stop.

I think the acceptability of the paper is contingent on whether the tuning of alpha is considered a sufficiently significant contribution. The authors themselves noted that their method (alpha = 1) is similar to HRL---differing only in the introduction of a termination signal. This in and of itself suggests that the main contribution of the paper boils down to learning a suitable choice of alpha to manage the termination signal.

I would also like to better understand the distinction between the author’s method versus HRL with Stop. Both methods have employ a low-level network capable of pre-emptive stopping. How, then, is the termination signal for HRL with Stop trained?

If the authors can convincingly demonstrate the novelty of the proposal to learn the terminal signal via extrinsic reward supervision, and if the other reviewers feel similarly convinced, then I would feel more comfortable re-evaluating my concerns about the significance of this work.

I would also, in general, encourage a more thoughtful exposition of the results shown in Table 1. Can the authors posit/explain why, for example, High-Level Only performs so much better on AS than the other models, but so poorly on the other metrics?

**Experience Assessment:**

I do not know much about this area.

**Review Assessment: Checking Correctness Of Derivations And Theory:**

I did not assess the derivations or theory.

**Review Assessment: Checking Correctness Of Experiments:**

I assessed the sensibility of the experiments.

**Review Assessment: Thoroughness In Paper Reading:**

I read the paper at least twice and used my best judgement in assessing the paper.

---

> ### Author Response · Authors · 2019-11-14
> **Response to Reviewer #2**
>
> Thank you for the constructive comments. We address the concerns as below,
> 1. We would like to reiterate the contribution of our work:  we propose a general hierarchical RL framework that 1) builds upon the human specified sub-goal space to incorporate human prior knowledge, that makes such a challenge task easier to learn and the solution interpretable; 2) optimizes with the intrinsic and the extrinsic rewards jointly to overcome the lingering inconsistency coming from the unsatisfied prior knowledge.
>     Existing work either learn to build the hierarchy and the sub-goal space themselves (such as Option-Critic method), or fully adopt the human-specified sub-goal space without considering the possibility that the prior knowledge may not be accurate enough for learning the hierarchical policy (such as HRL method). Both of them can be seen as specific cases of our framework (setting alpha as 0 or 1). However, our empirical experiments show neither one performs as well as our method, demonstrating that not only the value of the alpha matters, but also the way to properly integrate the advantages of both methods is of high significance.
>     Compared to HRL with stop method, though we both have a termination signal, HRL with stop method simply extends the HRL method by adding an additional “stop” action into the low-level atomic action space. Therefore, training the “stop” action remains the same as other low-level actions, i.e. $\theta_{\pi_l} \gets \theta_{\pi_l} + \nabla_{\theta_{\pi_l}} \log\pi_{\theta_{\pi_l}}(a|s, g, sg)(Q^i_l(s, g, sg, a)-V^i_l(s, g, sg))$, which depends on the intrinsic advantage the “stop” action brings compared to other low-level actions. While our method trains the termination signal by $ \theta_t \gets \theta_t - \nabla_{\theta_t}term_{\theta_t}(s, g, sg)(Q^e_h(s, g, sg)-V^e_h(s, g))$, that depends on the extrinsic advantage the current sub-goal brings compared to other sub-goals. The experimental results also demonstrate our training strategy is more efficient.
>
> 2. For the experimental results, we define AS metric as the average steps over all successful cases. As we can see from Table 1, High-level only achieves very low success rate (SR), so the small AS actually indicates the method can successfully reach the goal states only when the starting positions are close to the goal states. Since SPL and AR take both SR and AS into consideration, they are reasonably low for the High-level only method.

---

### Official Review · AnonReviewer1 · 2019-10-31
**Official Blind Review #1**

**Rating:** 3

**Review:**


------------------------------------------------------------------------------------
Rebuttal Response:
Thanks for the clarifications. Nevertheless, the rebuttal and the comments of the other reviewers did not convince me that this paper is ready for publication at ICLR and I keep my vote with weak reject. IMO this paper can be improved by either focussing more on the HRL part and performing simpler qualitative evaluations to highlight the HRL contribution OR by focussing completly on the robotics part by incorporating more classical robotics approaches and demonstrating their shortcommings within the experiments.

------------------------------------------------------------------------------------
Summary:
The paper proposes a hierarchical reinforcement learning scheme to search for objects specified by an image. The proposed learning approach is applied to a virtual house setting and compared against multiple baselines.

I like that the authors do an extensive comparison of different baselines and compare their results. My main concern is the setup with the task, which seems quite artificial. Learning to search for objects using pure RL seems like neglecting all robotics research from the past 50 years. By now we can generate maps, planners and low-level control policies to navigate within these maps. Such approaches would be able to remember the objects location and just return to them and do not need to discover them 1000x times to remember them. Therefore, one would only need to learn an optimal search pattern. Therefore, I would like to see the proposed HRL approach in a more appropriate experiment or even more excitingly be combined with classical robotics. I think that this combination should be quite exciting.

Regarding the HRL, could the authors please state their contributions in more detail? There is quite some work on subgoal generation within HRL. How does your work differ from these?

Currently, I am for borderline reject but I am happy to increase my rating during the rebuttal, when the authors clarify the motivation for the experiment and their contribution to HRL.

Minor Comments:
I think that the acronym is badly chosen. The term ROS is already famously coined within the robotics community for the Robot Operating System. Therefore, using this acronym for a robotics tasks is really confusing.


**Experience Assessment:**

I have read many papers in this area.

**Review Assessment: Checking Correctness Of Derivations And Theory:**

I assessed the sensibility of the derivations and theory.

**Review Assessment: Checking Correctness Of Experiments:**

I assessed the sensibility of the experiments.

**Review Assessment: Thoroughness In Paper Reading:**

I made a quick assessment of this paper.

---

> ### Author Response · Authors · 2019-11-14
> **Response to Reviewer #1**
>
> Thank you for the constructive comments. We address the concerns as below.
> 1. We definitely agree with the reviewer that the classic robotic research, such as map-based navigation, has achieved great success, while we also believe they still have limits in the robotic object search task and we are trying to explore a new way to overcome these limits. To be specific ,
>     1) Map is not always necessary for target-driven navigation. Building a map is non-trivial and consequently navigating with the map may be inefficient especially when the indoor environment is typically dynamic.
>     2) Separating the mapping and planning is again not necessary and may compromise the robustness of the whole system.
>     3) To search the target object specified by an image, the robot needs an object recognition model. Combining the object recognition model with the classic navigation algorithm may require additional efforts since they are not designed to work together.
>
> 2. Regarding the HRL, we would like to reiterate the contribution of our work: our proposed hierarchical RL 1) builds upon the human specified sub-goal space to incorporate human prior knowledge, that makes such a challenge task easier to learn and the solution interpretable; 2) optimizes with the intrinsic and the extrinsic rewards jointly to overcome the lingering inconsistency coming from the unsatisfied prior knowledge.
>   Although there are many work on sub-goal generation, they either learn to build the hierarchy and the sub-goal space themselves without prior knowledge, leaving the performance on challenge tasks much to be desired (as the poor performance of the Option-Critic method in our experiments demonstrated), or they fully adopt the human-specified sub-goal space without considering the possibility that the prior knowledge may not be accurate enough for learning the hierarchical policy. In fact, the more challenging the task is, the more difficult it is for humans to provide satisfied prior knowledge and designate sub-goal space.
>
> 3. Thanks again for the suggestions on the ROS acronym, we will come up with a more proper one.

---

### Decision · Program_Chairs · 2019-12-19

**Decision:**

Reject

**Comment:**

This paper introduces a two-level hierarchical reinforcement learning approach, applied to the problem of a robot searching for an object specified by an image.  The system incorporates a human-specified subgoal space, and learns low-level policies that balance the intrinsic and extrinsic rewards.  The method is tested in simulations against several baselines.

The reviewer discussion highlighted strengths and weaknesses of the paper.  One strength is the extensive comparisons with alternative approaches on this task.  The main weakness is the paper did not adequately distinguish between which aspects of the system were generic to HRL and which aspects are particular to robot object search.  The paper was not general enough to be understood as a generic HRL method. It was also ignoring much relevant background knowledge (robot mapping and navigation) if the paper is intended to be primarily about robot object search.  The paper did not convince the reviewers that the proposed method was desirable for either hierarchical reinforcement learning or for robot object search.

This paper is not ready for publication as the contribution was not sufficiently clear to the readers.